



# The *fortedata* R package: open-science datasets from a manipulative experiment testing forest resilience

Jeff W. Atkins[1], Elizabeth Agee[2], Alexandra Barry[1], Kyla M. Dahlin[3], Kalyn Dorheim[4], Maxim S. Grigri[1], Lisa T. Haber[1], Laura J. Hickey[1], Aaron G. Kamoske[3], Kayla Mathes[1], Catherine McGuigan[1], Evan Paris[5], Stephanie C. Pennington[4], Carly Rodriguez[6], Autym Shafer[1], Alexey Shiklomanov[7], Jason Tallant[8], Christopher M. Gough[1], Ben Bond-Lamberty[4]

[1]Department of Biology, Virginia Commonwealth University, Richmond, VA 23059, USA
[2]Environmental Sciences Division and Climate Change Science Institute, Oak Ridge National Laboratory, Oak Ridge, TN, 37831, USA
[3]Department of Geography, Environment, & Spatial Sciences, Michigan State University, East Lansing, MI 48824, USA
[4]Joint Global Change Research Institute at the University of Maryland, Pacific Northwest National Laboratory, College Park, MD, 20740, USA
[5]Vassar College, Poughkeepsie, NY, 12604, USA
[6] Western Colorado University, Gunnison, CO, 81231, USA
[7] NASA Goddard Space Flight Center, Greenbelt, MD 20771, USA
[8]University of Michigan Biological Station, Pellston, MI, 49769, USA

*Correspondence to*: Jeff W. Atkins (jwatkins6@vcu.edu)

**Abstract.**

The *fortedata* R package is an open data notebook from the Forest Resilience Threshold Experiment (FoRTE)--a modeling and manipulative field experiment that tests the effects of disturbance severity and disturbance type on carbon cycling dynamics in a temperate forest. Package data consists of measurements of carbon pools and fluxes and ancillary measurements to help analyse and users analyse and interpret carbon cycling over time. Currently the package includes data and metadata from the first two years of FoRTE, and serves as a central, updatable resource for the FoRTE project team and is intended as a resource for external users over the course of the experiment and in perpetuity. Further, it supports all associated FoRTE publications, analyses, and modeling efforts. This increases efficiency, consistency, compatibility, and productivity, while minimizing duplicated effort and error propagation that can arise as a function of a large, distributed and collaborative effort. More broadly, *fortedata* represents an innovative, collaborative way of approaching science that unites and expedites the delivery of complementary datasets in near real time to the broader scientific community, increasing transparency and reproducibility of taxpayer-funded science. *fortedata* is available via GitHub: https://github.com/FoRTExperiment/fortedata and detailed documentation on the access, used, and applications of *fortedata* are available at: https://fortexperiment.github.io/fortedata/ The first public release, version 1.0.1 is also archived at: https://doi.org/10.5281/zenodo.3936146 (Atkins et al. 2020b). All level one data products are also available outside of the package as .csv files: https://doi.org/10.6084/m9.figshare.12292490.v3 (Atkins et al. 2020c).



## 1 Introduction

Disturbance alters multiple carbon (C) cycling processes and, as a result, may affect forest C uptake and storage (Williams et al. 2016). The magnitude, timing and duration of changes in the C cycle following disturbance vary among forests (Amiro et al. 2010, Luo and Weng 2011, Coomes et al. 2012, Hicke et al. 2012, Gough et al. 2013, Peters et al. 2013, Vanderwel et al. 2013, Flower and Gonzalez-Meler 2015; Gu et al. 2019). These responses may differ as a function of disturbance severity, type, and frequency along with the physical, structural, and biological properties of the affected ecosystem (Amiro et al., 2010; Williams et al., 2012; Scheuermann et al. 2018; Rebane et al. 2019; Fahey et al. 2020; Atkins et al. 2020a). Understanding which forest ecosystems are most vulnerable to disturbance and, conversely, what characteristics of an ecosystem confer C cycling stability, remains an important frontier crucial to forecasting changes in the terrestrial C sink in the face of rising global disturbance frequencies (Frelich and Reich, 1999; White and Jentsch, 2001; Johnstone et al. 2010; 2016). Large-scale manipulative experiments may be particularly useful to identify the C fluxes and drivers that determine ecosystem C balance following disturbance (Fahey et al., 2020; Gough et al., 2013; Shiels and González, 2014).

Honing the prediction of how forests respond to disturbance, however, requires the parallel examination of mechanisms leading to the stability or decline of multiple C stocks and fluxes to changing disturbance regimes. The calculation and interpretation of forest ecosystem C balance necessitates repeated measurements of aboveground C stocks and fluxes through tree and litterfall inventories and belowground processes including root production and soil respiration--the total $CO_2$ efflux from roots and microbes to the atmosphere. Complementary process and structural measurements such as leaf physiology, morphology and chemistry along with remotely sensed measures of canopy structure and physiology provide important ancillary data useful to the interpretation of changes in C fluxes following disturbance. Few comprehensive datasets from such experiments exist in the public domain, and those that exist are almost never published in near-real-time concurrently as an experiment is conducted, which limits testing hypotheses related to forest resilience and functional change beyond the focus of the project and slows the scientific enterprise more broadly (Falster et al. 2019).

The "open data" movement in science emphasizes transparency, reproducibility, and the moral imperative of making publicly funded research products broadly available (Culina et al. 2018). A specific example of this is 'open notebook' science where the entire data record of a research project is made publicly available in near-real time with the goal of generating, integrating, documenting, and reporting heterogeneous data streams (e,g. Bond-Lamberty et al. 2016; Falster et al. 2019). Open notebook science helps create accountability and transparency by documenting the provenance of research data from conceptualization



to publication, and fights against the 'file drawer' effect of lost data (Rosenthal 1979). The ability for a project team to pull

from one well-documented and consistent open data notebook increases research productivity and efficiency--streamlining the process of data curation and manipulation, and eliminating errors or inconsistencies that may otherwise be introduced from multiple copies of datasets across multiple workstations. In turn, this increases the potential for reproducibility and data use outside of a core project (Powers and Hampton, 2019; Schapira et al., 2019, Gallagher et al. 2020). Open data notebooks also perfectly complement mentoring and teaching—simultaneously serving to rapidly and effectively onboard new team members

to the project, while also providing project-based learning opportunities in the classroom that teach open science and data science skills.

The goal of this manuscript is to (i) describe the scientific context and goals of the Forest Resilience Threshold Experiment (FoRTE), (ii) describe its experimental design and high-level measurement protocols, and (iii) document the open-source

*fortedata* package that serves as the project data repository. The systematically documented and transparent approach to science outlined in this manuscript and in the *fortedata* package surpasses the data-sharing expectations of publishers and funding bodies—specifically the publication of data prior to manuscript(s) submission--and may be considered as a model for future experiments and projects that is in line with widely-adopted principles concerning the management and stewardship of scientific data (See FAIR Principles, Wilkinson et al. 2016).


## 2 The FoRTE Project

FoRTE is a modeling and manipulative experiment that aims to identify the mechanisms underlying C cycling response to disturbance—specifically net primary productivity (NPP) resilience and its decline following disturbance. It centers on a manipulative field experiment located in northern, lower Michigan at the University of Michigan Biological Station (45.58

N, 84.71 W) with experimental plots that span ~8 ha of regionally representative landforms and forest types (Fig. 1). Data from the field experiment also informs a series of modeling experiments; specifically, data included in this package is used to initialize, calibrate, and validate dynamic vegetation model simulations of forest function and its responses to disturbance (e.g. Shiklomanov et al. *In Press*).

The experimental design follows a hierarchical structure with four replicates (A, B, C, D) of each factorial combination of disturbance severity (4 levels) and type (2 levels) (Fig. 1a, 1b). Within each replicate, each 0.5 ha plot was randomly assigned a disturbance severity level of 0, 45, 65, or 85% gross defoliation, respectively (Fig 1a). Each plot is bisected, with each half subjected to a disturbance treatment preferentially targeting large (top-down) or small (bottom-up) canopy trees (Fig. 1). All trees larger than 8 cm in diameter-at-breast-height (DBH) are classified as canopy trees. An intensively surveyed 0.1 ha subplot

is nested within each disturbance severity-treatment combination—there are a total of 32 subplots (Fig. 1). The standard nomenclature for subplots is a concatenation of the replicate (A, B, C, D) plot number (01, 02, 03, 04) and subplot location (E

for east side of the plot, or W for the west side of the plot) referred to in datasets by the variable name of *subplot_id* (Fig. 1b; Table S1-S3). Within each subplot, all canopy trees are measured (DBH) and geolocated (Total no. of measured trees 3165; Fig. 2) and terrestrial laser scans using both 2D and 3D lidar (light detection and ranging) are taken to estimate canopy structural

traits (Atkins et al. 2018; Fahey et al. 2019).

Within each subplot, a series of C cycling and environmental measurements are taken at nested subplots. There are two types of nested subplot: 1) nested subplots 0, 1, 3, 5, 7 are 1 $m^2$ plots located at plot center (0) and 10 m off plot center at cardinal directions (1 = north, 3 = east, 5 = south, and 7 = west) (Fig. 1b) where environmental measurements such as soil volumetric

water content, soil temperature, soil $CO_2$ efflux, and hemispherical imagery are taken; 2) nested subplots 2, 4, 6, and 8 are 4 $m^2$ vegetation survey plots located 8 m from plot center at intercardinal directions (2 = northeast, 4 = southeast, 6 = southwest, and 8 = northwest)(Fig. 1b) where understory leaf physiology, morphology, and chemistry measurements are taken. Additionally, all stems in the $4m^2$ vegetation survey plots, including those below the 8 cm DBH canopy threshold, are counted and identified to the species level. The data detailed above are meant to be illustrative, but not entirely inclusive, of what is

being measured in FoRTE. Additional environmental measurements will be taken as FoRTE matures and then added to *fortedata* in as near-real time as possible—including, but not limited to, soil chemical and physical properties, dendrometer readings, canopy profiles from 3D terrestrial lidar, fine root production, root density profiles, and data products from a NEON Airborne Observation Platform fly-over from 2019.

**3.1 The *fortedata* Package**

*fortedata* is a package for the R language (R Core Team, 2020) that includes field data from FoRTE. *fortedata* version 1.0.1 (Atkins et al. 2020b) includes: leaf physiology, canopy structural traits, soil respiration, litterfall, soil micrometeorology, and forest inventory data for the years 2018 and 2019.

**3.2 Versioning and Archiving**

*fortedata* uses semantic versioning (https://semver.org/), meaning version numbering follows an "x.y.z" format where x is the major version number, y the minor version number, and z is the patch version number. For example, this manuscript specifically details version 1.0.0. The major version number (x) only changes when there is a major change in overall package structure or there is expansive update in data—for example, following the inclusion of all data for a given field season. The

minor version number (y) changes follow less notable changes, such as minor changes in functionality or the addition of minor data products. Changes in the patch version number (z) represent minor bug fixes or error corrections that do not affect package structure. Following each (major) release a DOI will be issued and the data archived by Zenodo (https://zenodo.org/). All

changes to data and code are immediately available through the GitHub repository, but only official releases will be issued a DOI.

**3.3 Package License**

*fortedata* is under a CC-BY-4 license (https://creativecommons.org/licenses/by/4.0/); see the "LICENSE file in the repository. This is identical to that used by e.g. Ameriflux and FLUXNET Tier 1. This license provides that users may copy and redistribute this R package and its associated data in any medium or format, adapting and building upon them for any scientific or commercial purpose, as long as appropriate credit is given. We request that users cite this manuscript (see Section 3.4), and

strongly encourage them to (i) cite all constituent dataset primary publications (see *fd_publications()*, and (ii) involve data contributors as co-authors when possible and appropriate.

**3.4 Citing the FoRTE Data Package**

Papers or other research products using any FoRTE data should cite both this publication and the *fortedata* package, including

the package version used. Appropriate citations can be found via the command *citation("fortedata").*

**4.1 Using the FoRTE Data Package to Access FoRTE Data**

It is necessary to install and use the *fortedata* R package in order to access FoRTE data. *fortedata* can be installed directly from GitHub (https://github.com/FoRTExperiment/fortedata) (Atkins et al. 2020b) using the *devtools* package in R (Wickham

et al. 2020):

*devtools::install_github("FoRTExperiment/fortedata")*
*library(fortedata)*

We plan to submit *fortedata* to the Comprehensive R Archive Network (CRAN), the common clearing house for all

standardized R software packages.

**4.2 FoRTE Data Package Structure**

The package is structured as a collection of independent datasets with standardized plot notation, date (ISO 8601 standard YYYY-MM-DD) and time (HH:MM:SS TZ) formatting (see *fd_plot_metadata* and Tables S1-S3 for more information). Datasets are available via user-facing, external functions outlined below. Additional metadata, instrument specifications, and





abbreviated measurement protocols are available in supplementary information (Tables S1-S11) and in package documentation. Currently available functions: include:

\* *fd_inventory()* returns a single dataset of the forest inventory data, including diameter-at-breast height (DBH), latitude, longitude, species, as well as information on vitality and canopy position (Table S4). There are 3165 observations, all measured

in 2018 (Fig. 2, 3). DBH measurements were taken with a Haglof PDII Digital Caliper (Haglof, Inc., Madison, MS, USA). Longitude and latitude were measured using a Trimble R1 GNSS Receiver (Trimble; Sunnyvale, CA, USA) which has an accuracy range of +/- 30 cm.

\* *fd_soil_respiration()* returns a single dataset of 2780 observations each of soil $CO_2$ efflux ($\mu$mol $CO_2$ m$^{-2}$ s$^{-1}$), soil temperature (°C; integrated from 0 to 7 cm depth), and volumetric water content (%) for the year 2019 (Figs. 2, 4; Table S5). Soil $CO_2$

efflux was measured using a LI-6400 XT (LI-COR Biosciences; Lincoln, NE) with a soil $CO_2$ flux chamber model 6400-09 attachment with a measurement accuracy of +/- 5 $\mu$mol mol$^{-1}$ maximum deviation. Soil temperature was measured using the attached soil temperature probe, with an accuracy of +/- 1.5 °C. Soil moisture was measured using a Campbell HS2 HydroSense II time domain reflectometer (Campbell Scientific; Logan, UT, USA) with a measurement accuracy of +/- 3% and accurate range of 0 – 50%.

\* *fd_leaf_spectrometry()* returns a single dataset of vegetation indices derived from leaf-level spectrometry data collected via a CI-710 handheld spectrometer (Table S6). The dataset includes 7155 observations of spectral indices for three species each in eight subplots within the D replicate (Figs. 1 and 2).

\* *fd_photosynthesis()* returns a single dataset of leaf physiology variables, including photosynthesis, transpiration, etc, measured using a LI-6400 XT (LI-COR Biosciences; Lincoln, NE)(Table S7) with a measurement accuracy of +/- 5 $\mu$mol mol$^{-}$

$^{1}$ maximum deviation.. The dataset includes 2215 observations from 2018 (Fig. 2).

\* *fd_litter()* returns a single dataset of litter mass collected via litter traps (four in each subplot, at nested sampling points 1, 3, 5, 7). The data include the tare + oven-dried mass of leaves as well as the tare weight (the empty bag), by species, by subplot (Table S7). The data also include notations for "CWD", the collection of coarse woody debris (e.g. sticks, branches), and "MIX", fragments of leaves too small to identify to the species levels as well as other missed organic fragments in the basket.

Litter mass can be calculated by subtracting the tare weight from the mass + tare. There are a total of 340 observations included in the dataset from 2018 (Fig. 5).

\* *fd_hemi_camera()* returns a single dataset that includes derived estimates of leaf area index, gap fraction, clumping index, and NDVI (normalized difference vegetation index) from terrestrial, upward-facing hemispherical photos looking into the forest canopy taken 1 meter above-ground (Table S9). The dataset includes 1028 observations of each variable from 2018 and

2019 (Fig. 2).

\* *fd_canopy_structure()* returns a single dataset that includes estimates of canopy structural traits such as height, area/density, openness, complexity, and arrangement derived from terrestrial lidar and processed using *forestr* version 1.0.1 (Atkins et al. 2018) in R Version 3.6.2 (R Core Team, 2020). The package includes 62 observations for each metric (28 canopy structural



metrics are included in *forestr* v1.0.1 that estimate canopy structural traits such as area/density, openness, arrangement,
heterogeneity, and layering (Atkins et al. 2018; Fahey et al. 2019)) from 2018 (Table S10).

\* *fd_ceptometer()* returns a single dataset that includes estimates of the fraction of photosynthetically available radiation
(faPAR) absorbed by the canopy as well as leaf area index (LAI)--each derived from a handheld ceptometer (LP-80; Decagon
Devices)(Table S11) with a resolution of 1 μmol m$^{-2}$ s$^{-1}$ and accuracy of +/- 5%. The dataset includes 32 observations of each
variable from 2019 and 16 from 2018 (Fig. 2).


Brief summaries of certain datasets are available via summary functions:

\* *fd_inventory_summary()* returns a summary of the *fd_inventory()* dataset that includes stocking density (in stems per ha) and
mean basal area (m2 per ha) averaged at the subplot level (n = 32) grouped by Replicate, Plot, and Subplot variables.

**4.3 Accessing FoRTE Data without Using *fortedata***

All data contained in *fortedata* can also be accessed directly via FigShare (https://doi.org/10.6084/m9.figshare.12292490.v3,
Atkins et al. 2020c) as a compressed file containing all output generated from each function in *fortedata* (Atkins et al. 2020c).
This mirror of the dataset will be updated with each major release of *fortedata*.

**4.4 FoRTE Documentation and Vignettes**

This manuscript serves as the primary documentation for FoRTE data and all code to reproduce this manuscript—including
the tables and plots herein—are available in the package (https://github.com/FoRTExperiment/fortedata/tree/master/essd). The
package also includes additional supporting documentation via R's standard help system. Vignettes, which are guided tutorials
that include example code or background information such as experimental design and proposal narratives, are also included
both in the package and online and can be accessed via *BrowseVignettes("fortedata")*. Vignettes are currently available for
the functions above and additional vignettes will be added as new data products are incorporated into fortedata."

Supporting project information, including detailed methods and data collection information (introduced briefly below and in
Supplementary Information Tables S1-S11), can be found within package documentation: function help files (e.g.
*?fd_inventory()*) and package vignettes--which can be accessed via *browseVignettes("fortedata")* or online:
https://fortexperiment.github.io/fortedata/. The funded project narrative (NSF DEB-165509) can be accessed directly within
in the package via *vignette("fd_forte_proposal_vignette")* and outlines hypotheses, objectives, proposed methods, and
supporting literature for the project.

**4.5 Testing and Quality Assurance**

The *fortedata* R package has a wide variety of unit tests that test code functionality, typically via assertions about function behavior, but also by verifying behavior of those functions when importing datasets. As datasets within *fortedata* differ in composition and format, they may create a variety of errors. Unit tests, detailed below, ensure that entries in these datasets are realistic and valid. These tests are run automatically every time *fortedata* code or data is updated on GitHub, ensuring
continuing package validity for end users. These tests include error checks on:

- Appropriate date and timestamp formatting
- Data class verification (e.g. plot numbers as integer values, soil $CO_2$ efflux measurements as numeric, etc.)
- Out of bounds latitude or longitude values
- Appropriately formatted plot metadata that adheres to FoRTE naming conventions
- Out of bound values (e.g. unreasonable, unrealistic, erroneous entries) for environmental measurements (e.g. negative values for tree DBH, soil water content $< 0$ or $>100$, etc.).

The appropriate method of uncertainty quantification for any given dataset herein *fortedata* may vary based on the use,
application, or analyses of these data. To this end, we have provided extensive documentation for end users to make these calculations based on their own judgement, discretion, or discipline specific needs. This is why there is no direct quantification of uncertainty for datasets contained within *fortedata*. These data are "raw" and represent unmodified point measurements, taken according to each instrument's or method's standards. Any uncertainties associated with measurements, either instrument or method specific, are detailed above in section 4.2 and in tables S5-S11.

**4.6 Reporting Issues**

We use the *fortedata* GitHub issue tracker (https://github.com/FoRTEExperiment/fortedata/issues ) to track and categorize user improvement suggestions, problems or errors with the R package code and included data, as well as requests for new variables or functionality, and/or other questions. All past and current issues are viewable to the public, and new issues can be contributed by anyone with a (free) GitHub account.


**5 Conclusion**

The lack of existing publicly available datasets comprehensively documenting forest and ecosystem manipulations limits our ability to test hypotheses related to forest resilience and functional change, broadly. While projects such as FoRTE push our boundaries of understanding the mechanisms that facilitate resilience, the additional effort to make the project as open and
transparent as possible, including making project data available in near real-time, increases the impact of the project. FoRTE

and the *fortedata* package serve as one model for future experiments and projects by showcasing the advantages of supplying centralized project data openly and in near-real time to investigators within and external to the project. This approach is above and beyond the typical requirements and expectations for data availability, particularly in field-based–ecology where standard conventions for data availability, if and where they do exist, call for reporting only upon project completion or publication.

The results of such modular practices often limit data availability to single spreadsheets of varying quality with limited, sometimes non-existent, metadata. We argue that open-notebook science should be the new science normal, whenever possible—when we fail to provide timely, open, and usable data, we fall short of our duty as scientists, and in doing so jeopardize scientific advancement and its societal benefits:

*"...the free, open, and responsible practice of science is fundamental to scientific advancement for both human and environmental well-being. Science requires freedom of movement, collaboration, and communication, as well as equitable access to data and resources. It requires scientists to conduct and communicate scientific work for the benefit of society, with excellence, integrity, respect, fairness, trustworthiness, clarity, and transparency."* – "The Responsibilities and Rights of Scientists", American Geophysical Union, 2017


We do acknowledge there may be legitimate barriers for some scientists/project teams—such as limited access to reliable internet, to resources to acquire necessary computational skills, to budgeted time, to supportive and collaborative environments where open-science is rewarded—these challenges require our attention and support. In addition, some types of data (proprietary, human-subject) clearly require different standards and practices. That said, where there exists the privilege of

having access to the necessary resources to conduct science openly and equitably, choosing to do otherwise is unconscionable. Open science approaches should be the rule, not the exception and we anticipate that the release of *fortedata* in near-real time will motivate external collaboration, facilitate data exchange within the project, and provide project-wide data transparency, consistency, and availability, as well as increased team member efficiency and productivity.

**Data Availability**

*fortedata* is available via GitHub (https://github.com/FoRTExperiment/fortedata) and can be installed and accessed directly within the R programming language as outlined above. Additionally, the first version of *fortedata* (version 1.0.1) outlined in this paper is archived at: https://doi.org/10.5281/zenodo.3936146 (Atkin et al. 2020b). We have also made all level one data products accessible as formatted .csv files with accompanying documentation available via Figshare:

https://doi.org/10.6084/m9.figshare.12292490.v3 (Atkins et al. 2020c).

**Author Contributions**

JWA, BBL, and CMG wrote the manuscript. JWA, EAA, AB, KMD, LTH, LJH, MSG, AGK, KM, CM, EP, CR, AS, and JT collected, processed, and provisioned data for the package, including providing details on methods and metadata included



within the manuscript and in package documentation. JWA, BBL, KD, SCP, and AS contributed code and package oversight. BBL and CMG envisioned, proposed, and oversaw the project. All authors contributed to manuscript discussion, editing, and revision.

**Acknowledgements**

FoRTE is funded by the National Science Foundation [DEB-1655095]. NEON AOP data collection was supported by NSF award #1702379 to KMD. CR, LJH, and EP, whole or in part, were supported by the UMBS REU Program (NSF #1659338) We'd also like the gratefully acknowledge the University of Michigan Biological Station for their continued support.

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

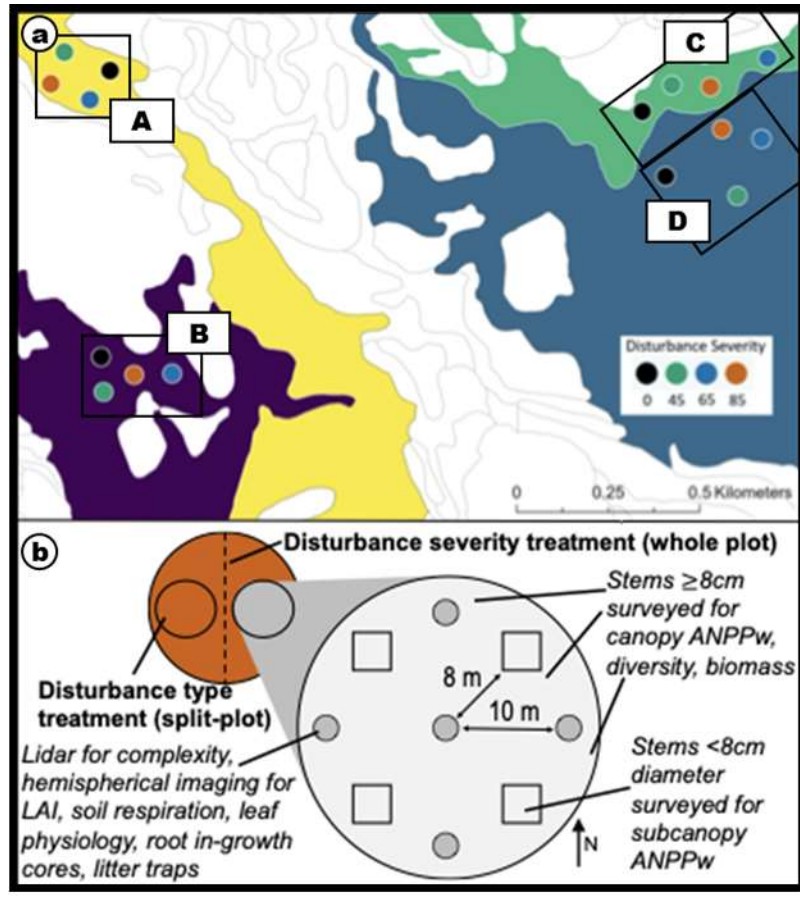


**Figure 1: (a) Map showing the distribution of plots in relation to landform types (*)—colors indicate assigned severity levels. Plot replicates are grouped (A, B, C, D); (b) Subplot diagram showing position of nested subplots for sampling and arrangement of subplots within the plot (orange).**





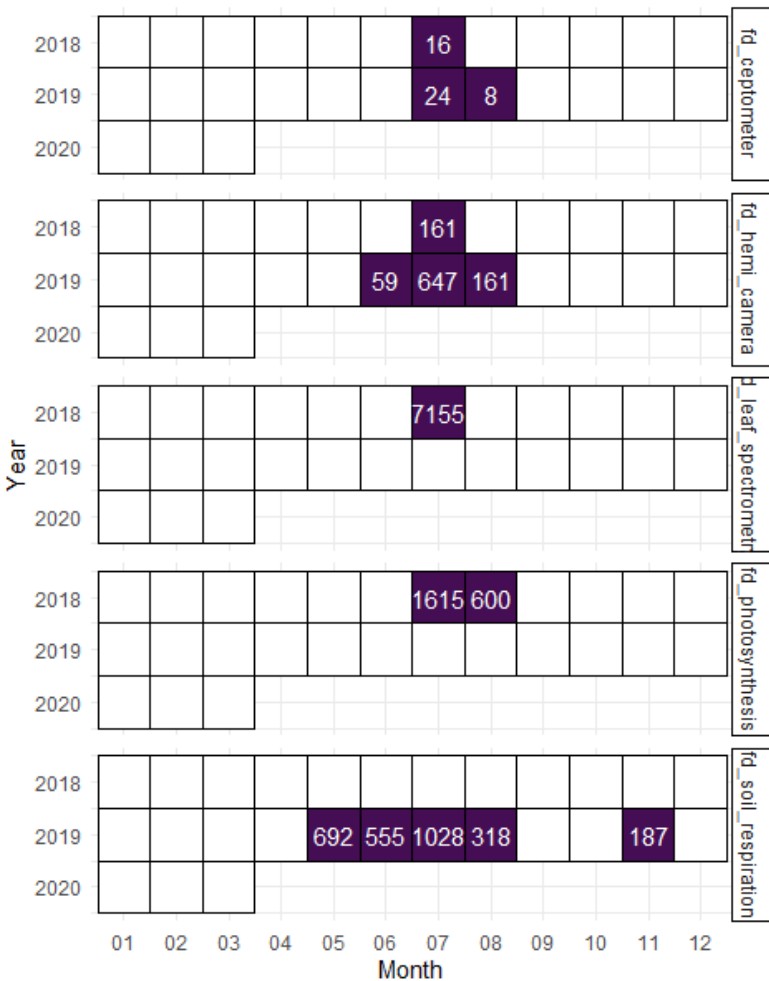

**Figure 2: Number of available records as of March 30, 2020 for time-series datasets including light availability from *fd_ceptometer()*, camera derived LAI from *fd_hemi_camera*, leaf-level vegetation spectra indices from *fd_spectrometry*, photosynthesis and stomatal conductance from *fd_photosynthesis*, and soil CO₂ efflux from *fd_soil_respiration*.**


**Figure 3: Diameter-at-breast height (DBH) distributions for each species, grouped by replicate. The bounds of each box in the boxplot represents the 25th percentile at the lower bound, and the 75th percentile at the upper, and the horizontal line is the median. Lines extending from the lower and upper bounds represent values that are 1.5 times the interquartile range for the minimum and maximum values, respectively, while black circles indicate outliers. Above each box plot, *n* is the number of observations.**

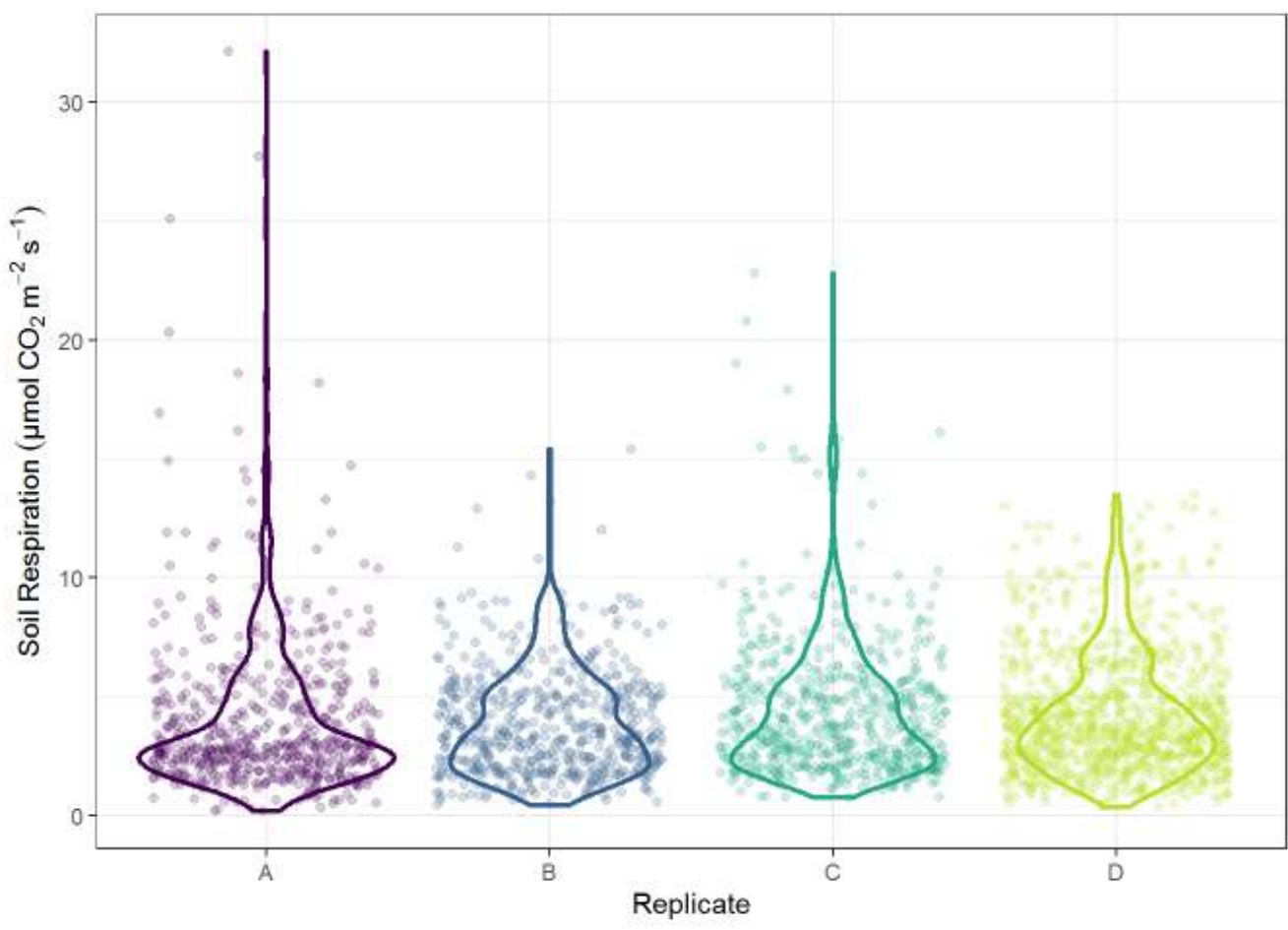


**Figure 4: Distribution of soil CO₂ Efflux values from May-November, 2019 by replicate. Lines represent distribution while points are individual measures.**



**Figure 5. Distribution of litter mass values for 2018 by replicate. Lines represent distribution while points are individual**
**measures.**