# Peer review of "The *fortedata* R package: open-science datasets from a manipulative experiment testing forest resilience"

_Earth System Science Data, 2020_

## Referee Comment (RC1) · Joseph Stachelek (Referee) · 27 Oct 2020

**1   General comments**

The authors of this paper have done an excellent job following the principles of open data. They have quite thoroughly documented their data and made it readily available in a variety of archives. A real highlight of their efforts is the use of R package testing infrastructure for data validation. I also like how the authors have used minimal software dependencies in their package.  This is something that likely increases the project's long-term value.

[Figure]

**2   Specific comments**

Will the data size ever become unmanageble to ship with the package? The authors say they plan to submit the package to CRAN but the current package size ( 9mb) is larger than the CRAN limit of 5mb.

What is the planned data update frequency of fortedata? The authors say it will be updated as near-real time as possible. Is this likely to be annual, semi-annual, or some longer timestep?

L129: What is the approximate field season for the project?

I wish measurement units were embedded for all data columns. I see this is the case for `fd_invetory()$dbh_cm` but not `fd_soil_respiration$soil_co2_efflux`.

L211: A nice-to-have but certainly not required feature, a machine readable representation of the metadata associated with fortedata beyond the human readable manuscript.

Figures 3 and 5 are not referenced in the text.

Might want to add a URL and BugReports field to the DESCRIPTION file

**3   Technical corrections**

Abstract: I suggest using a different term for "level one data product" in the abstract or explaining it.

L24: ancillary measurements to help analyse and users analyse and interpret carbon cycling -> ancillary measurements to help users analyse and interpret carbon cycling

L43: This seems like an excessive number of citations for the preceding statement.

---

## Referee Comment (RC2) · Anonymous Referee #2 · 12 Nov 2020

**General comments:**

The fortedata R package is an easy to use platform to access the FoRTE project datasets, which are logically organized and of general value. I have no reservations that the package will be useful as the project matures and see no issues with its main functionality. Nevertheless, I have four main areas of critique to bring the package to publication quality: (1) the limited documentation within the R package, (2) the usefulness of the vignettes, (3) the lack of clarity on present and future data availability, and (4) individual comments broken down by dataset under the "Specific comments" header below.

For publication, the package should include additional annotation, description of data, and the fleshing out of package documentation within R. As an example, the fd\_inventory() help file lacks description other than "Raw inventory table" and a note on collection using a Haglof Postex Inventory Unit. A few extra steps do get the user to the metadata table with a description of variables, but I see value in fleshing out the description here to the level done in the data paper. There are also notes like that in the help file for fd\_canopy\_structure\_summary() that simply says "For now this is pretty basic." And, there are instances like in the help file for fd\_subplots() where the listed columns don't match those of the dataset. Some documentation is missing from the submitted ESSD preprint and SI. There are functions in the package that are not described such as forte\_colors() and plot\_metadata(). Pointing out these issues is not to nitpick these particular details, but I think these small details add up to make the package feel in development rather than a final product for outside users.

Similarly, the vignettes (https://fortexperiment.github.io/fortedata/articles/index.html) are promising and needed to familiarize outside users with the package and data but only two of seven vignettes plot or manipulate the data. The other vignettes are descriptive of the project or just show the datasets being called. Vignettes are not needed to show the dataset being called. Rather, a handful of more illustrative uses of the package would be better as vignettes. Even the more fleshed out vignette (https://fortexperiment.github.io/fortedata/articles/fd\_inventory\_vignette.html) only shows a density plot by replicate. Again, the vignette doesn't really help the user parse the dataset much since there is no breakdown by species, subplot, etc. Certainly the group has made more interesting and complete visualizations/analyses that can be shown like your Figure 3. These vignettes don't help the outside user get a head start on analyzing these data.

Information on anticipated release schedules for new data (and for which variables) is needed, or at least a statement about where that information can be found once known. Will additional sites be added? What additional data will be released for the
years 2019 and 2020? The selling point of near real-time data availability through the R package is compelling, but Figure 2 does not convince that there is real-time data being released (e.g., no spectrometry or photosynthesis data since mid-2018). Are these data to be made available following separate publication or do they not exist? As a naïve user, I need to know if what I am analyzing is complete. The R package is still a suitable platform to distribute these data but these details are necessary to make the data publicly usable and not just publicly available.

Specific comments:

The following are comments broken down by dataset: fd\_inventory: Explain/note data with missing date information, species codes marked ???? (unidentifiable species?) for DBH inventory. What is the column "tag" in the fd\_inventory data set? It lacks a description in the SI. This appears to maybe be an index column but there are a few errors in the numbering. "tag" 2236-2244 have an erroneous 9 in front of them, it appears. Should be DP II instead of PD II caliper, presumably.

fd\_soil\_respiration: 2,791 observations are in the dataset when loaded through the R package. Again there are missing timestamp values (1,622 or over half the data, which even if the date is available is notable).

fd\_leaf\_spectrometry: Was this dataset exported as only the head of the data? Only 8 rows of data import using the R package when 7,155 are expected. There is an additional column "tree\_id" that is not in the format of USDA PLANTS species codes and not defined under Table S6 in the SI.

fd\_litter: Is "MISC" code equivalent to "MIX" as defined in L184 or is it actually Mikania scandens? What are the codes "SWD" and "FAGRE", these don't appear to be USDA PLANTS codes? Is there a reason to not just use a column of the actual litter mass rather than the intermediate columns for bag mass and bag+litter mass?

fd\_hemi\_camera: Again a mismatch between reported observations and the number
in the R package dataset.

fd\_canopy\_structure: Again a mismatch between reported observations and the number in the R package dataset. In the associated Table S10 variables are separated by periods instead of underscores as in the actual data and the other SI tables. There are additional undescribed variables such as the skew and kurtosis intensity missing from Table S10.

L158: fd\_plot\_metadata() is not how the function appears in the R package. The help file in the R package does not describe how to use it to get the metadata properly.

fd\_metadata(table = "fd\_inventory") returns the whole metadata tibble.

Technical corrections: No major issues found.

---

## Author Comment (AC1) · 14 Dec 2020

On behalf of myself and the other authors, I would like to express my appreciation for the thoughtful and detailed reviews from both Dr. Stachelek and Reviewer 2. I have incorporated their feedback into both the revised manuscript and the fortedata package as annotated below.

I have made significant changes to the package and manuscript including expanding the amount of data included, adding updates via GitHub about future data availability, expansion and harmonization of package vignettes, adding additional in-package documentation (e.g. function descriptions), and cleaning up language for clarity in the

[Figure]

manuscript. The function 'fd_mortality' has also been added, which provides additional information to 'fd_inventory' about which trees were stem-girdled.

In response to Dr. Stachelek"s question about CRAN, while it is my intention to post this package for consideration to CRAN,I do acknowledge that it currently exceeds the maximum package size of 5 mb and this requires attention. The bulk of the package file size is composed of vignette files, specifically images and plots. Careful consideration will be taken to navigate between limiting the file size of the package, while maximizing the ease of use/instructional components of the package. All options will be explored.

In regards to data update frequency, additional information has been provided in the GitHub package readme including recent updates, current progress on data sets undergoing QA/QC. Currently the package includes canopy structural data for 2018-2020, leaf phys data for 2018 and 2020, soil respiration, soil temp, and soil water for 2019-2020. By the end of 2020, leaf photosynthesis data (currently 2018 only) for 2018 - 2020 will be included as will canopy light interception data. The QA/QC of some data has been hindered by COVID-19 (e.g. hemispherical imagery, 3-D lidar metrics). Also, see below in response to reviewer 2 for additional elaboration. The following lines 124-126 have been changed to address R1's specific comment on the project field season as well as issues mentioned above:

". . . leaf physiology, canopy structural traits, soil respiration, litterfall, soil micrometeorology, and forest inventory data for the years 2018, 2019, and 2020. Additional project data and data products will be incorporated over the lifetime of the project (initial FoRTE NSF funding 2018-2022)."

Response to specific comments from Dr. Stachelek:

"I wish measurement units were embedded for all data columns. I see this is the case for fd_invetory()$dbh_cm but not fd_soil_respiration$soil_co2_efflux." – reviewer comment

[Figure]

I have made it more clear in the function help sections and vignettes which units are being used in which data set, but given that some column names might balloon in size tremendously if this convention were always followed, I have opted to retain the current slimmer structure.

"L211: A nice-to-have but certainly not required feature, a machine readable representation of the metadata associated with fortedata beyond the human readable manuscript."

I agree and will pursue this in further package updates.

"Figures 3 and 5 are not referenced in the text." – reviewer comment

This has been corrected. Figure 3 is referenced in ∼line 165 in the fd_inventory description and figure 5 is referenced at ∼line 185 in the fd_litter description.

"Might want to add a URL and BugReports field to the DESCRIPTION file" – reviewer comment

These have been added.

Abstract: I suggest using a different term for "level one data product" in the abstract or explaining it. – reviewer comment

That term has been removed.

L24: ancillary measurements to help analyse and users analyse and interpret carbon cycling -> ancillary measurements to help users analyse and interpret carbon cycling – reviewer comment

Changed.

L43: This seems like an excessive number of citations for the preceding statement.

I feel strongly this helps contextualize the FoRTE project as these references individually touch on different aspects or outcomes of forest and ecological disturbance relative

to FoRTE. And given the myriad data streams being collected at FoRTE, I feel broad contextualization is helpful.

Responding to the 2nd reviewer's comments, specific reporting on data availability has been added to each data set's corresponding vignette. This is accomplished through the inclusion of a dynamic waffle chart that reports the number of records by the month and year for each data function. (an extension of Figure 2 from the main ESSD manuscript). These vignettes are rebuilt when data become available in the package. Further the readme.md file, which is the front page of the GitHub repository now includes a section on version updates that includes information on when data are incorporated and future data availability and timelines (when possible).

Many data sets have been updated with 2019 and 2020 data, including soil respiration, soil temperature, soil moisture, and canopy structure/lidar data sets. The mortality data set, accessed via 'fd_mortality' includes the forest inventory data set with the associated experimental treatment data–including which trees were stem-girdled and their associated metadata. Updating the hemispherical imagery, leaf litter, and light availability data sets have been delayed due to COVID-19. Additionally, other ancillary data (e.g. forest dendrometer readings and met data) are in the pipeline for 2021.

We have also sought to address related concerns in text as well–changing the language of the manuscript to largely avoid the term "near-real-time". We have changed lines 29 - 31 in the manuscript to:

"More broadly, fortedata represents an innovative, collaborative way of approaching science that unites and expedites the delivery of complementary datasets in near real time to the broader scientific community, increasing transparency and reproducibility of taxpayer-funded science."

Lines 115-116 have been changed to:

"Additional environmental measurements will be taken as FoRTE matures and then

added to fortedata in prior to incorporation in conventional data products such as research papers near-real time as possible—including, but not limited to, soil chemical and physical properties, dendrometer readings, canopy profiles from 3D terrestrial lidar, fine root production, root density profiles, and data products from a NEON Airborne Observation Platform 2019 fly-over. The fortedata readme file includes updates on the progress of current and future data availability."

Lines 254-258 have been changed to:

" . . .additional effort to make the project as open and transparent as possible, including the expeditious delivery of project data making project data available in near real-time, increases the impact of the project . . . the fortedata package serves as one model for future experiments and projects by showcasing the advantages of supplying centralized project data openly and in near-real time to investigators within and external to the project."

We have made the following major changes:

- Vignettes have been expanded and standardized. Now function vignettes follow the pattern of introduction with provided scientific background, followed by a description of the current available data, function descriptions, supporting field methods, and references. Additionally, an Examples vignette which shows how to do higher level analysis with some FoRTE data has been added and will be expanded in future releases.

Specific comments:

"The following are comments broken down by dataset: fd_inventory: Explain/note data with missing date information, species codes marked ???? (unidentifiable species?) for DBH inventory. What is the column "tag" in the fd_inventory data set? It lacks a description in the SI. This appears to maybe be an index column but there are a few errors in the numbering. "tag" 2236-2244 have an erroneous 9 in front of them, it appears. Should be DP II instead of PD II caliper, presumably." – reviewer comment

I have adjusted the inventory data set as such. The leading 9's are now gone, the tag number along with species info is more detailed in ?fd_inventory(). And yes ???? is an unidentifiable species. There are plans in 2022 to remeasure DBH. Work this next season will help id any trees marked as ???? or that have any other issues mentioned in notes.

fd_soil_respiration: 2,791 observations are in the dataset when loaded through the R package. Again there are missing timestamp values (1,622 or over half the data, which even if the date is available is notable). – reviewer comment

Corrected in text.

fd_leaf_spectrometry: Was this dataset exported as only the head of the data? Only 8 rows of data import using the R package when 7,155 are expected. There is an additional column "tree_id" that is not in the format of USDA PLANTS species codes and not defined under Table S6 in the SI. – reviewer comment

I have made significant changes to this data set. It now has leaf_id and tree_id data columns, which correspond to specific leaves on specific trees where canopy spectrometer and photosynthesis measurements were taken–hence the absence of USDA codes. This has been added to the ?fd_leaf_spectrometry. The strange data filter issuing has also been solved. I have expanded the ?fd_leaf_spectrometer too to be more useful and pointed to the vignette which is also expanded.

fd_litter: Is "MISC" code equivalent to "MIX" as defined in L184 or is it actually Mikania scandens? What are the codes "SWD" and "FAGRE", these don't appear to be USDA PLANTS codes? Is there a reason to not just use a column of the actual litter mass rather than the intermediate columns for bag mass and bag+litter mass? – reviewer comment

I have added the column 'fraction' which details if each sample is "leaves" or something else...specifically "misc" which are small unidentifiable organic fragments. Or "fwd"

which is fine woody debris such as sticks and twigs. All of these pools are important for carbon accounting even if only leaves are used for the leaf area estimates. The ?fd_litter has been expanded accordingly.

fd_hemi_camera: Again a mismatch between reported observations and the number – reviewer comment

Corrected

fd_canopy_structure: Again a mismatch between reported observations and the number in the R package dataset. In the associated Table S10 variables are separated by periods instead of underscores as in the actual data and the other SI tables. There are additional undescribed variables such as the skew and kurtosis intensity missing from Table S10. – reviewer comment

This has been corrected.

L158: fd_plot_metadata() is not how the function appears in the R package. The help file in the R package does not describe how to use it to get the metadata properly. – reviewer comment

This has been added to package vignettes and information and included in the manuscript.

fd_metadata(table = "fd_inventory") returns the whole metadata tibble. – reviewer comment

This is a table that is used in the package self-build to populate manuscript and information tables. It is also used internally in project testing to check columns and units. To your point, you can import what you need using the code filter(fd_metadata, table == "fd_inventory")

–

Thank you for your consideration.

Dr. Jeff Atkins et al.